# Human mobility and outbreak origins in epidemic spread: Insights from agent-based modeling

Konstantin A. Klochkov[1,2*], Ivan E. Kozlov[1], Elena N. Ilina[1], Alexander I. Manolov[1]

**1** Research Institute for Systems Biology and Medicine, Moscow, Russia, **2** Moscow Center for Advanced Studies, Moscow, Russia

\* konstantin.a.klochkov@gmail.com

## Abstract

Human mobility is a key driver of early epidemic spread, and restricting travel remains one of the principal non-pharmaceutical interventions. To better understand how infections propagate through real-world mobility networks, it is essential to disaggregate their components and characterize the functional relationships between mobility flows and epidemic metrics. Here we introduce transCovasim, an agent-based extension of Covasim that explicitly couples parallel city simulations via inter-city traveler exchange while preserving the initial social network structure within each population, enabling controlled experiments on mobility and disease dynamics. Using transCovasim, we analyze a two-city system with equal or unequal populations and a hub-and-satellite commuting network parameterized to a Moscow-like setting. In paired identical cities, the mean lag between epidemic peaks scales approximately linearly with the logarithm of inter-city traffic, with steeper delays at lower transmissibility; epidemic variability declines as flows increase. With unequal city sizes, mobility primarily redistributes infections between cities; first-order Sobol' indices show that peak magnitude is largely insensitive to city's flows of different directions when sizes are comparable, while sensitivity for a smaller city shifts toward flow from a bigger city as the asymmetry increases. In the hub-and-satellite network, reducing commuting flows before the peak significantly lowers peak incidence, and cumulative infections can still be reduced when restrictions are introduced after the peak; early 100-fold cuts outperform 10-fold cuts, but produce similar results when introduced into the late exponential phase. Finally, dynamic time warping applied to surveillance curves identifies the outbreak's origin: under Moscow-like flows, accuracy reaches 78% by day 50 with 10% daily testing, and approaches 100% at lower connectivity. These results clarify how specific mobility patterns shape epidemic timing and burden and provide actionable guidance for mobility-targeted non-pharmaceutical interventions and early source attribution.

**Data availability statement:** All data and code used for running experiments and visualizing results, and the transCovasim source code are available online at https://github.com/KonstantinKlochkovv/transportation-motifs. transCovasim provides a framework for studying human mobility in agent-based models, allowing agents to be exchanged between parallel simulations of the base agent-based model while preserving the initial social network structure within each population.

**Funding:** This work was supported by a subsidy from Rospotrebnadzor (The Federal Service for Surveillance on Consumer Rights Protection and Human Wellbeing), No. 141-02-2023-208 (to KAK, IEK, ENI, and AIM). The funders had no role in study design, data collection and analysis, decision to publish, or preparation of the manuscript.

**Competing interests:** The authors have declared that no competing interests exist.

## Author summary

Human mobility governs how epidemics spread between cities, yet policy often treats it as a single lever. We introduce transCovasim, an agent-based extension of Covasim that links parallel city simulations via explicit traveler exchange while preserving the initial social network structure within each population, allowing controlled tests of pairwise inter-city traffic and hub-and-satellite commuting. In identical cities, the lag between epidemic peaks shrinks approximately linearly with the logarithm of traffic, and stochastic variability decreases as flows rise, especially at lower transmissibility. With unequal sizes, mobility chiefly redistributes infections; sensitivity increases as asymmetry grows. Cutting commuter traffic before the peak reliably reduces peak incidence, while later cuts still lower cumulative burden; early deeper cuts help more. Comparing surveillance curves with time-series alignment can identify the likely source within about a month under moderate testing. These results provide quantitative, network-aware guidance for targeting connections and timing interventions.

## Introduction

Human mobility plays a key role in the early spread of epidemics, and restricting mobility is an important non-pharmaceutical intervention when vaccines or therapeutics are unavailable. High-resolution surveillance during the COVID-19 pandemic provided extensive evidence for the contribution of human mobility to epidemic dissemination at both national and global scales.

A novel coronavirus outbreak (COVID-19) emerged during the Spring Festival travel season (Chunyun), a period when several billion trips occur across China [1]. Stringent restrictions on outbound travel from Wuhan were imposed only about two weeks after the festival began; by then, the estimated probability of at least one imported case exceeded 0.5 in 130 Chinese cities, 107 of which had reported cases by January 26, 2020 [1]. At that point, the risk of importation into Europe, where many countries had yet to detect cases, was already very high [2,3], although travel curbs reduced international importations from China by roughly 80% by mid-February [4].

COVID-19 initially spread preferentially to cities with large populations and high tourist volumes from Wuhan [5]. During the first wave in mainland China, population mobility from Wuhan was strongly associated with transmission: both cumulative and incident case counts correlated with aggregate traffic, especially rail and bus, toward other cities [6–9]. Case numbers in Hubei correlated most strongly with road and rail trips from Wuhan, whereas outside Hubei they correlated more with rail and air travel. Moreover, node centrality metrics in road and air networks (e.g., strength, degree, PageRank) explained cumulative case counts better than direct inflow from Wuhan; for road networks, local centrality (strength, degree) outperformed both flow measures and global centrality (PageRank), consistent with predominantly short-range spread via roads. Epidemic arrival times were most strongly associated with

the shortest effective distance from Wuhan [10] in railway networks, somewhat less in road networks, and least in air networks [6]. Taken together, early spread in China was driven primarily by high-volume land transport (rail and bus).

Simulations of the COVID-19 epidemic in England and Wales, launched with different outbreak origins, demonstrate similar qualitative behavior: wave-like spread from urban areas to rural areas [11] (a similar result for measles was obtained by Grenfell et al. [12]). In Europe, five different clusters of temporal dependencies of excess mortality on time during the COVID-19 period are identified, which are associated with specific geographical regions (for example, a difference between western and eastern Europe is noted) [13].

Using global commercial flight data, Stenseth et al. [14] showed that, for the first two months from January 11, 2020, modelled international spread of COVID-19 closely tracked observations until widespread interventions were introduced. In their analysis, passenger reduction was more effective than entry quarantine, although both were less effective than reducing transmissibility [14]. Complementary modelling for China estimated that non-pharmaceutical interventions, including Wuhan travel ban, delayed infection arrival in other cities by a mean of 2.91 days; transport restrictions alone reduced cumulative cases by day 50 nearly fourfold, and by almost 25-fold when combined with other measures [5]. Notably, however, the authors found no clear evidence that restricting traffic outside Hubei reduced case numbers.

Human mobility interventions have proven effective when applied early in regions distant from the initial outbreak of the COVID-19 epidemic. Similar restrictions could have reduced the number of deaths in the islands macro-region of Italy by 12% if introduced almost a month earlier than they were, and by about 4% without early application of travel restrictions [15]. According to various estimates, international movement restrictions reduced the number of exported cases by 80% until mid-February 2020 [3,4,16]. On the other hand, mobility interventions did not show the same significant effect in mainland China, where the Wuhan travel ban delayed the epidemic progression by only 3–5 days [4,7], and in Iran, where no significant effect on COVID-19 trends was observed as a result of a smart travel ban policy without other active measures [17]. For practical policy purposes, it is important to prioritize limiting long-distance connections [6,18], implement restrictions early in the epidemic [15,19], and combine mobility interventions with community-level non-pharmaceutical interventions [4,17]. At the same time, it is necessary to weigh these epidemiological benefits against decline in health service use such as observed in the Democratic Republic of Congo during the COVID-19 pandemic [20].

Another epidemic in which the role of human mobility has been systematically examined is influenza.

Pairwise correlations in weekly influenza phases decline with inter-settlement distance and increase with population size; phase correlations also exhibit a nonlinear rise with traffic volume that plateaus at high flow levels [21]. Commuting flows promote synchronization of local epidemics, whereas long-range air travel largely governs international arrival times of pathogens [22]. September international air-traffic volume predicts the timing of the influenza-mortality peak, and the post-9/11 mobility downturn coincided with a delayed season [23,24]. By contrast, during the 2009 U.S. epidemic, inferred hubs of spread were not the most connected or densely populated areas [25].

Outbreaks seeded in high-density districts (>1,000 persons per $km^2$) exhibit substantially less predictable trajectories than those seeded in low-density areas for Ebola, influenza, and measles [26]. In metapopulation models with a generic pathogen and uniform transmissibility and recovery across patches, system stability (eventual fade-out) is governed by average commuting flow rather than by migration per se, although migration can still reshape spatial spread [27]. More broadly, mobility structures transmission, and correlations between inbound and outbound flows can modulate stability conditions [27].

The spread of infection in transport networks is described by a variety of different models, but conceptually they can be divided into the following groups: compartmental, network-based, and agent-based models. Compartmental models describe the number of people in different states (i.e., compartments, such as susceptible, infected, recovered) using differential equations for each individual settlement. To account for mobility, terms describing the movement of people in different states between cities and the incidence of disease during movement are added to these equations. Such models assume uniform mixing within each population and do not take into account heterogeneity in human behavior.

Network-based models represent settlements as network nodes and movements between them as edges, where infection can spread both within nodes and through edges (i.e., population-based models, e.g., reaction-diffusion equation describing the metapopulation system). Individual-based network-based models represent individuals as network nodes, connected in dense social networks within settlements, and movements between them as edges between these networks. In agent-based models, agents represent individuals that have their own characteristics (e.g., age) and individual disease progression, can infect each other according to a social network, perform various actions and move between different locations [28].

When transitioning from compartmental models to network-based models and from network-based models to agent-based models, we gain the ability to model heterogeneity in human behavior and more complex interactions between people, but this also increases the complexity of the model and the data requirements. In existing agent-based model studies, the task of investigating the impact of human mobility on the spread of epidemics has been addressed using specific, non-general-purpose models or using highly detailed models such as EpiSimdemics [29] and EMOD. This fact points to an existing gap in research and the need for an agent-based model with a balanced level of detail and complexity.

We aim to fill this gap with our model transCovasim, an agent-based epidemiological model built on Covasim [30]. Using this framework, we quantified how the shift in peak infection day between two cities depends on inter-city travel intensity and pathogen transmissibility; assessed sensitivity indices for flows of different directions and insensitivity index. We further characterized how system metrics change under alternative mobility interventions in hub-and-satellite networks and proposed a method to identify the outbreak's origin by exploiting temporal offsets between city-level epidemic curves.

## Results

### Two-city transport model with equal agent counts

We considered a two-city model with identical populations (100,000) and symmetric inter-city mobility (equal flows in different directions). A schematic of the setup is shown in Fig 1A. We ran 150 epidemic simulations seeded in one city, varying the daily inter-city traffic (fraction of the population per day) and pathogen infectiousness (fraction of the wild-type SARS-CoV-2 infectiousness).

The peak-time offset of the destination city relative to the origin city decreased monotonically with both infectiousness and inter-city traffic, and the result variance declined as well (Fig 1C, S1 Fig).

Fig 1B depicts the mean peak-day offset as a function of traffic flow across infectiousness levels. The offset is well described by a linear relationship with the logarithm of traffic flow, with slope magnitude decreasing as infectiousness increases (coefficient of determination $R^2 \in [0.98, 0.99]$). The t-statistic comparing peak days between the two cities is a non-monotonic, upward-convex function of traffic flow, attaining its maximum at flows below $10^{-4}$. This peak likely reflects discreteness effects when the transported cohort comprises only 1–10 agents. The t-statistic increases systematically with infectiousness (S1 Fig).

### Two-city transport model with unequal populations

We analyzed a two-city model with a fixed total population of 200,000 and varying population ratios (Fig 2A,B). The city of origin had a minimum size of 10,000 agents. In this regime, epidemic metrics depend only on flow from the neighboring city: both cumulative and peak incidence in the origin city increase with inter-city traffic, while they decrease in the neighbor. The peak day in the origin city is essentially invariant to flow, whereas the neighbor's peak day declines with flow and is nonlinear. Notably, changes in cumulative and peak incidence are of similar magnitude in both cities, indicating that mobility primarily redistributes infections between them (Fig 2A,C).

At the opposite extreme (origin city size 190,000 agents), the pattern reverses: metrics depend only on flow from the origin, leading to a redistribution of infections toward the neighboring city as traffic increases (Fig 2B,D).

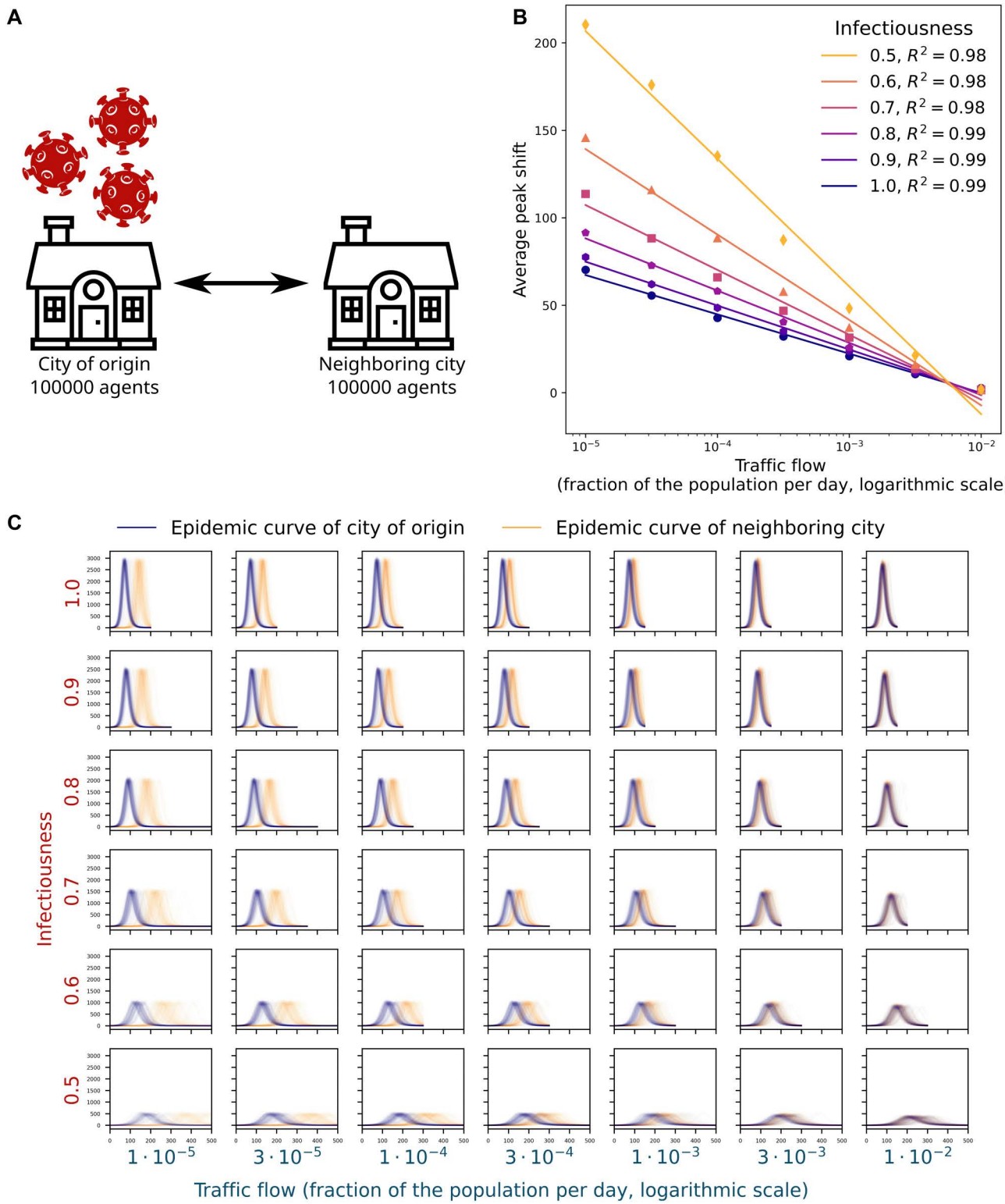

**Fig 1. Epidemic spread in a two-city system with identical populations.** A: Schematic of the transport model with symmetric inter-city flows. B: Mean offset in the peak-infection day of the destination city relative to the origin city as a function of inter-city traffic (fraction of the population per day; logarithmic scale) across multiple levels of infectiousness (fraction of wild-type SARS-CoV-2 infectiousness). C: Epidemic curves for the origin city (blue)

and the destination city (orange) for varying traffic (columns) and infectiousness (rows). Each panel summarizes 150 simulations with outliers removed. House and virus icons used in panel A are sourced from SVG Repo and Openclipart and are available under the CC0 1.0 Public Domain Dedication.

We computed first-order Sobol' sensitivity indices [31,32] for the inter-city flow from the outbreak origin and from the neighboring city, across three metrics, cumulative infections, peak incidence, and peak day, in both cities. Indices were estimated using a 4,096-sample Saltelli design [33], with traffic flows varied independently over $10^{-5} - 10^{-3}$.

Let us denote the first-order Sobol index as S and subtract all first-order Sobol indices from 1, calling this insensitivity and denoting as I ($I = 1 - \sum S_i$). In other words, insensitivity is the proportion of the dispersion of the results that is not associated with the variation of one of the transport flows alone.

As the population of the origin city increases, the first-order sensitivity to flow from the origin rises, reaching S = 0.82 for cumulative infections in the neighbor and S = 0.38 in the origin, while the sensitivity to flow from the neighbor declines from S = 0.91 (origin) and S = 0.63 (neighbor) to S < 0.1 for cumulative infections in both cities (Fig 2C,D,E). Thus, cumulative infections in both locations are governed primarily by the absolute number of travelers. For this metric, we observe the lowest insensitivity among those considered: sensitivity increases when city sizes (and hence absolute flows) are highly unequal, and decreases when flows are similar. Sensitivity is highest when the origin is smallest (I = 0.09 for the origin); when the origin is largest, the neighbor remains sensitive (I = 0.20), whereas the origin becomes comparatively insensitive (I = 0.63) (Fig 2E).

The maximum number of infections in a city is essentially independent of the tourist flow from that city (S < 0.2) for any population-size ratio. At the same time, the maximum number of infections in the origin city depends only on flow from the neighboring city, and only when its population is below 70,000 agents. For the neighboring city, the first-order sensitivity index with respect to flow from the origin increases as the origin's population grows (from S < 0.1 to S = 0.66) (Fig 2C,D,E). This metric is the most insensitive among those considered: when the two cities are similar in size, the insensitivity index exceeds 0.8. A small but noticeable sensitivity appears for the city with the smaller population when the population difference between cities is large (Fig 2E).

The peak infection day in the city of origin is largely insensitive to both flow from the origin and flow from the neighboring city (I > 0.9). For the neighboring city, the first-order sensitivity index with respect to flow from the origin increases as the origin's population grows (from S = 0.01 to S = 0.33), while the index with respect to the neighbor's own flow decreases (from S = 0.31 to S < 0.01) (Fig 2C,D,E). The neighboring city's peak-day insensitivity I remains within the range of 0.65–0.9 for all considered city size ratios (Fig 2E).

## Hub-and-satellite transport model

Another common motif in real transport systems is a large city surrounded by smaller, closely connected satellites. We therefore constructed a hub-and-satellite commuting network with one hub and four satellites. To approximate a realistic configuration, the hub-to-satellite population ratio matches that of Moscow and the Moscow region [34], while the total number of agents is set to one-twentieth of the real population to reduce computation time. Inter-satellite flows were generated using a gravity model [35–37], with traffic proportional to city populations and inversely proportional to the square of inter-city distance. A schematic of the resulting transport model is shown in Fig 3A.

We ran 150 simulations with the outbreak seeded in either a satellite or the hub (Fig 3A). We then introduced mobility interventions at various times: a 10-fold or 100-fold reduction of all flows in the network or only the flows directly connected to the origin city. For comparison, an additional 150 baseline simulations were run without restrictions. Outcomes were summarized by two metrics: cumulative infections and peak daily infections.

The left panels of Fig 3C,D show how the system metrics vary with the timing and type of intervention when the outbreak starts in the hub. The gray line indicates the mean (with standard deviation) from 150 simulations without

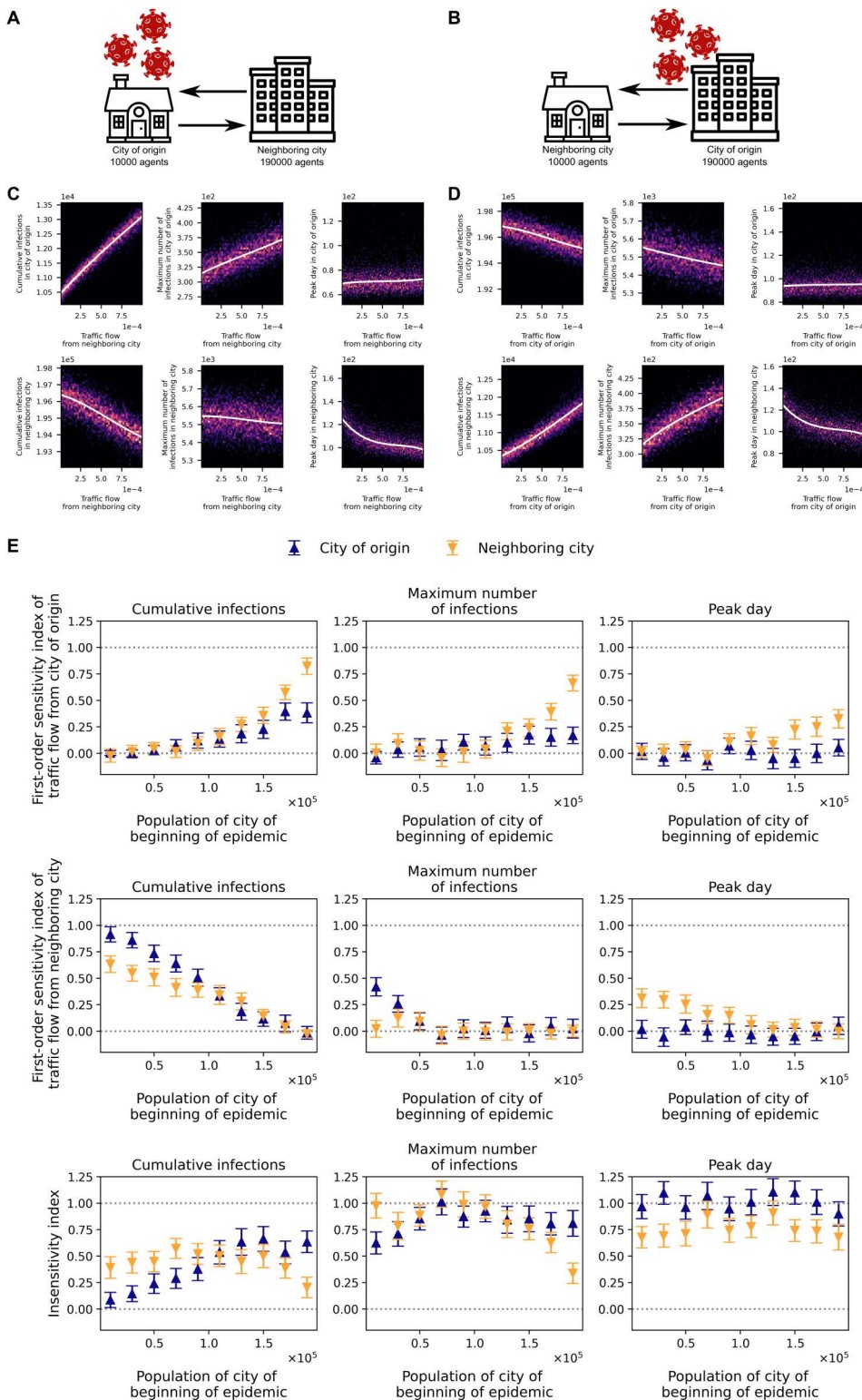

**Fig 2. Epidemic spread between two unequal-size cities.** A–B: Schematics of the two-city transport model with the outbreak seeded in a city of 10,000 agents (A) or 190,000 agents (B). C–D: Heatmaps of cumulative infections, peak daily infections, and peak-day timing (columns) for the origin city (top rows) and the neighboring city (bottom rows), as functions of (C) flow from the neighbor when the origin has 10,000 agents, and (D) flow from

the origin when it has 190,000 agents. E: First-order sensitivity indices versus the city-size ratio for the flow from the origin (top panels), the flow from the neighbor (middle panels), and the resulting insensitivity index (bottom panels), shown for cumulative infections, peak daily infections, and peak-day timing (columns) in the origin city (blue) and the neighbor (orange). Infectiousness is set to the wild-type SARS-CoV-2 level. House, apartment, and virus icons used in panels A and B are sourced from SVG Repo and Openclipart and are available under the CC0 1.0 Public Domain Dedication.

restrictions. An asterisk marks time points where the metric under restrictions differs significantly from the no-restriction baseline (Student's t-test, significance level 0.05 with multiple-comparisons adjustment). For context on intervention timing, Fig 3B plots the epidemic curves when the outbreak originates in the hub (left) or in a satellite (right).

When the outbreak begins in the hub, imposing restrictions up to the peak day yields a statistically significant reduction in peak infections (Fig 3D). Cumulative infections decline significantly even when restrictions are introduced after the peak and maintained through day 120, by which time daily new infections have fallen to about 14% of their maximum (Fig 3B,C). During days 40–70 (late exponential phase), the effects of 10-fold and 100-fold flow reductions are similar, whether applied network-wide or only on hub-satellite links, and the effect size remains essentially stable across the end of the exponential phase (Fig 3C,D). Even with interventions introduced at this stage, cumulative infections decrease by 3.8% ($p < 10^{-188}$) and peak infections by 8.5% ($p < 10^{-168}$) (Table 2). By contrast, the gap between 10-fold and 100-fold restrictions is most pronounced early in the epidemic: a 100-fold reduction lowers peak values by an additional factor of roughly 1.5–2 compared with interventions initiated at the end of the exponential phase (Table 1).

We observe analogous patterns when the outbreak starts in a satellite (right panels of Fig 3C,D; Tables 1, 2). The key difference is that restricting only the flows incident to that satellite is far less effective than network-wide restrictions, although both produce statistically significant reductions in the target metrics.

## Detecting the outbreak's origin in the hub-satellite model

Experiments were performed on the hub-satellite model described above, varying infectiousness and a global multiplier applied to all traffic flows. For each combination, 150 simulations were run; results are summarized in Fig 4. When the outbreak starts in the hub, satellite curves are markedly more synchronous than when the outbreak starts in a satellite, an effect evident even at a flow multiplier of 0.3. Subtle desynchronizations are more clearly revealed in phase portraits (Fig 4B). These observations motivate detecting the outbreak's origin via temporal dependencies between city-level trajectories.

To infer the outbreak's origin from temporal dependencies, we applied dynamic time warping (DTW) to the hub's incidence curve and each satellite's curve [38]. From the DTW alignment, we computed the mean temporal offset between each pair of curves by averaging the pointwise time shifts along the optimal path. The curve with the earliest inferred onset (smallest mean shift) was designated as the origin city; in the event of ties, one of the tied cities was selected at random (Fig 5A). To approximate real-world conditions, we incorporated agent testing (see Methods).

With Moscow-region-like flows, testing more than 2% of the population yields about 60% origin-detection accuracy by day 25, after which accuracy plateaus due to high synchrony. With lower testing fractions, the maximum attainable accuracy falls to roughly 45%. The best performance in this scenario is about 78% when testing 10% of the population, reached by day 50 at the latest (Fig 5B).

For hub-satellite networks with weaker connectivity, the maximum detection accuracy approaches 100% across all tested sampling fractions. However, in the time-testing plane, the heatmap reveals a region where reliable detection is impossible because too few infections are identified at those epidemic times (Fig 5C, S2 Fig).

At lower infectiousness, the boundary of the heatmap's nondetection region shifts to later epidemic times; at infectiousness equal to 0.5 of the wild-type SARS-CoV-2 level, this boundary is delayed by roughly twofold.

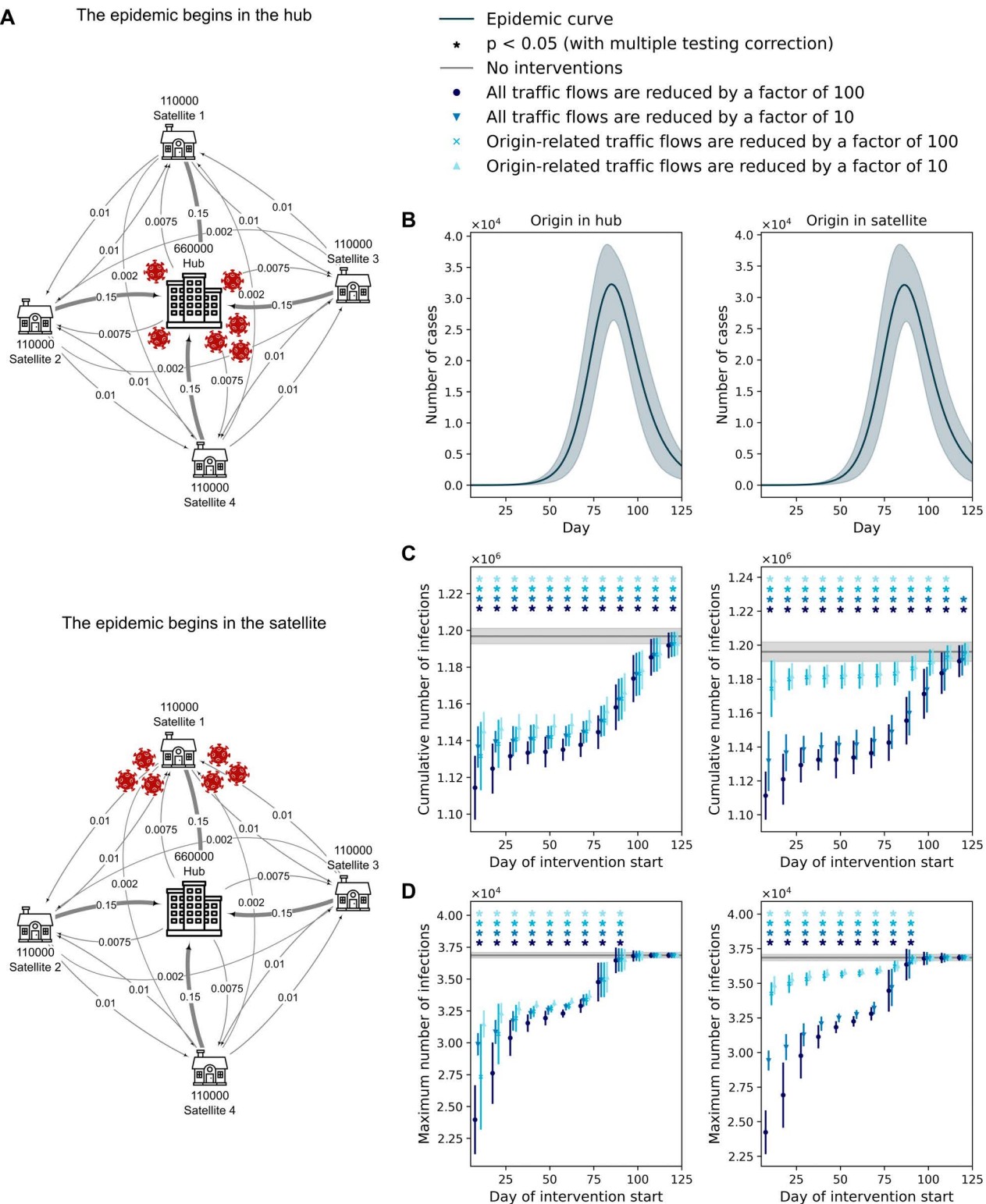

**Fig 3. Hub-and-satellite transport model.** A: Schematic of the hub-satellite system represented as a graph, with outbreaks initiated in the hub (top) or in a satellite (bottom). Vertex labels indicate city population sizes (number of agents), and edge labels indicate the daily commuting fraction of the population. B: Epidemic curves for outbreaks starting in the hub (left) or in a satellite (right). Shaded areas denote the standard deviation across 150

simulations. C–D: Cumulative infections (C) and peak infections (D) in the hub-satellite system as functions of the day interventions in commuting flows are introduced, for outbreaks starting in the hub (left) or in a satellite (right). Curves of different colors correspond to different transport-flow interventions. The gray curve indicates the mean across 150 baseline simulations without interventions; shaded areas show standard deviations. Stars mark statistically significant differences (Student's t-test, $p < 0.05$ with multiple-testing correction). Infectiousness is set to the wild-type SARS-CoV-2 level. House, apartment, and virus icons used in panel A are sourced from SVG Repo and Openclipart and are available under the CC0 1.0 Public Domain Dedication.

**Table 1. The magnitude of the reduction in target metrics when introducing transport restrictions on the 10th day of the epidemic.**

| | Origin in hub | | | | Origin in satellite | | | |
|---|---|---|---|---|---|---|---|---|
| | Restriction on all flows | | Restriction on hub-related flows | | Restriction on all flows | | Restriction on satellite-related flows | |
| | 100-fold | 10-fold | 100-fold | 10-fold | 100-fold | 10-fold | 100-fold | 10-fold |
| Cumulative infections reduction | 6.9% | 5.0% | 5.4% | 4.3% | 7.1% | 5.4% | 1.8% | 1.4% |
| Maximum infections reduction | 35.0% | 18.9% | 25.9% | 14.6% | 34.2% | 20.2% | 7.1% | 5.4% |

**Table 2. The magnitude of the reduction in target metrics when introducing transport restrictions on the 70th day of the epidemic.**

| | Origin in hub | | | | Origin in satellite | | | |
|---|---|---|---|---|---|---|---|---|
| | Restriction on all flows | | Restriction on hub-related flows | | Restriction on all flows | | Restriction on satellite-related flows | |
| | 100-fold | 10-fold | 100-fold | 10-fold | 100-fold | 10-fold | 100-fold | 10-fold |
| Cumulative infections reduction | 4.9% | 4.4% | 4.3% | 3.8% | 5.0% | 4.4% | 1.2% | 1.1% |
| Maximum infections reduction | 10.8% | 9.6% | 9.4% | 8.5% | 11.0% | 9.9% | 2.7% | 2.4% |

## Conclusion and discussion

In this study, we analyzed two foundational transport-network configurations, the two-city system and the hub-satellite system, that underlie more complex mobility structures.

For two identical cities, we found a linear relationship between the neighbor's peak-time shift (relative to the outbreak city) and the logarithm of inter-city traffic, with the slope magnitude decreasing as transmissibility increases. This supports the use of logarithmic effective distances [6,10] when a single characteristic travel mode dominates, in our experiments, movements with mean duration 7 days, 40 contacts, and a transmissibility multiplier equal to that of random contacts in Covasim. In practice, this suggests such distances are appropriate on temporal or spatial scales where one mode prevails (e.g., air travel on an international scale, or rail during China's Spring Festival).

We also observed an upward-convex relationship between the t-statistic comparing peak days in the two cities and the inter-city traffic flow. The presence of an extremum suggests a discreteness effect at flows corresponding to roughly 1–10 travelers per 100,000 residents per day, where result variance increases markedly. Notably, the t-statistic's dependence on transmissibility is the inverse of the peak-shift trend: as transmissibility rises, the mean peak shift shrinks while its variability shrinks even more, yielding a larger t-statistic.

In the two-city system with unequal sizes, cumulative infections in each city become more sensitive to traffic flows as size asymmetry and passenger volumes increase, and they depend on the absolute number of travelers. By contrast, when cities are similar in size, the peak infections are largely independent of flows; with large size disparities, the smaller city's peak becomes flow-sensitive. The neighbor's peak day depends on the absolute number of travelers, whereas the origin's peak day is insensitive to changes in flows. Overall sensitivities are modest: for many city-size ratios, the first-order sensitivity to any single flow does not exceed 0.25, implying that no more than one quarter of the variance in a given metric is attributable to variation in that flow alone.

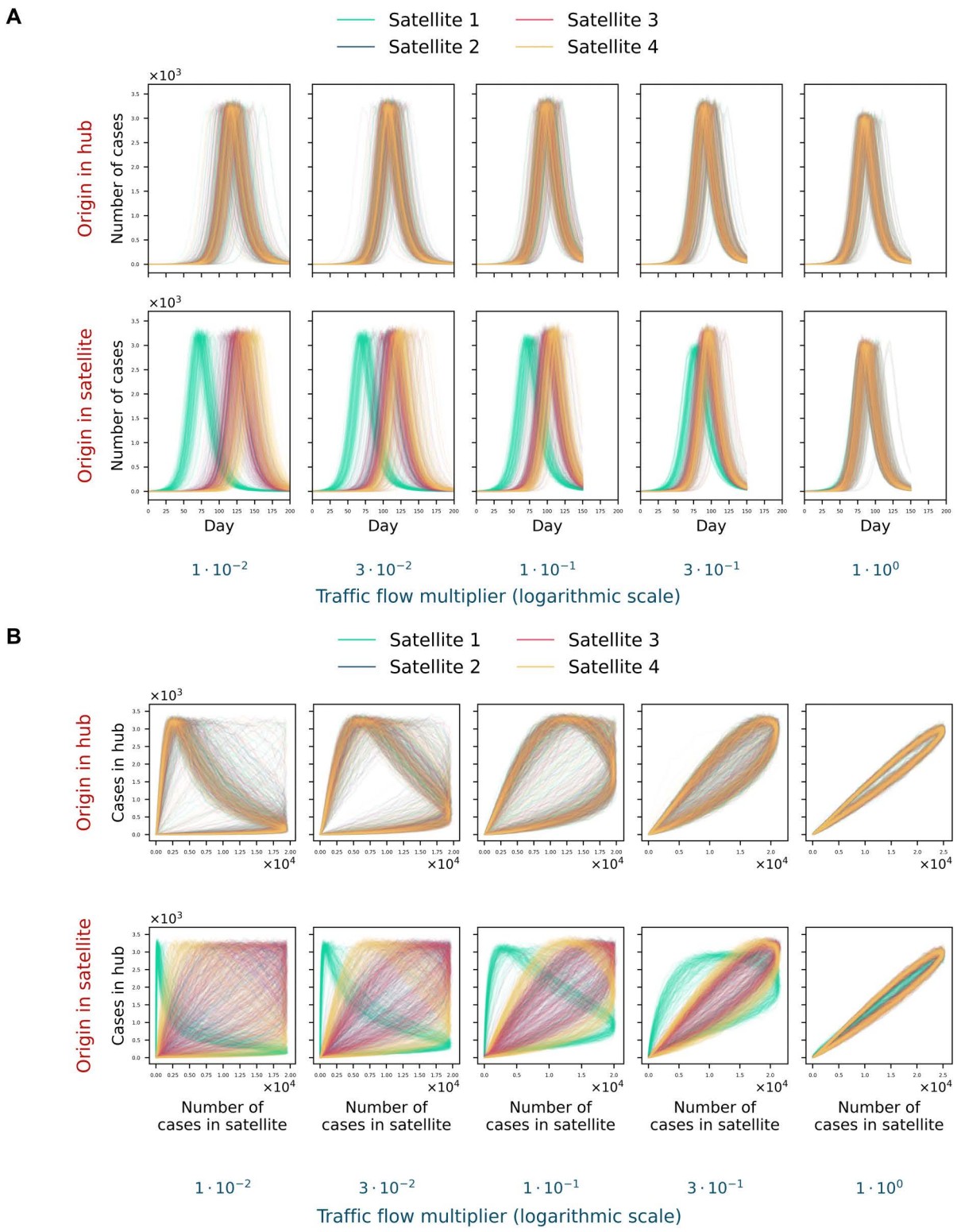

**Fig 4. Relative shifts among satellite epidemic curves in the hub-satellite system.** A–B: Satellite incidence curves (A) and phase portraits (B) for outbreaks seeded in the hub (top rows) or in a satellite (bottom rows), shown across traffic-flow multipliers (panel columns) with infectiousness set to the wild-type SARS-CoV-2 level. Phase portraits highlight temporal offsets that are less apparent in raw incidence curves.

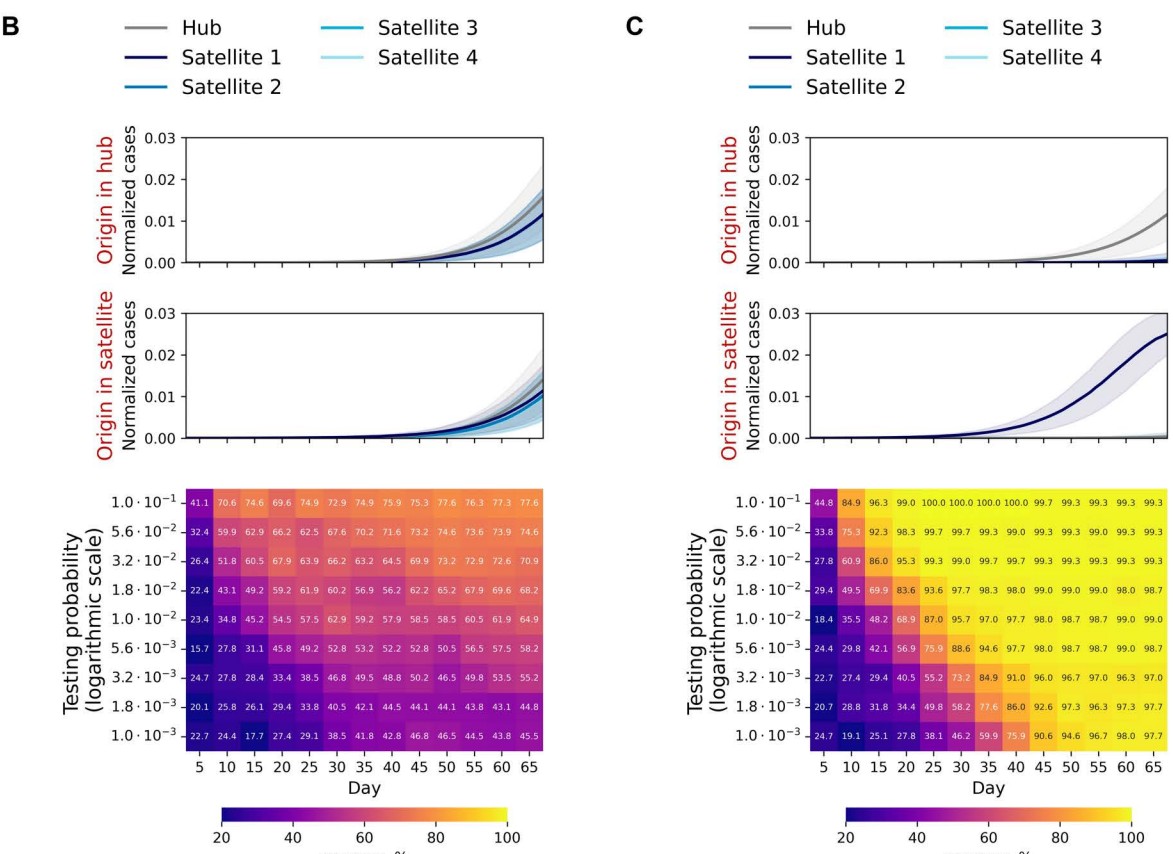

**Fig 5. Detecting the outbreak's origin in the hub-satellite model.** A: Schematic of the DTW-based method for identifying the origin city. B-C: Heatmaps of origin-detection accuracy versus epidemic day for systems with traffic flows equal to (B) or 100-fold smaller than (C) those of Moscow and the Moscow region. Testing is simulated by sampling detected infections from a binomial distribution.

In the hub-satellite system with an outbreak seeded in the hub, 10-fold and 100-fold reductions applied either network-wide or only on hub-linked flows yield statistically significant decreases in peak and cumulative infections when introduced by the peak and maintained through day 120, respectively. When implemented at the end of the exponential phase, all four restriction types perform similarly. By contrast, early in the epidemic, a 100-fold reduction achieves roughly 1.5–2 times greater peak reduction than a 10-fold reduction. Analogous patterns hold when the outbreak starts in a satellite, but restricting only satellite-related flows is notably less effective than network-wide measures. These results underscore the value of timely non-pharmaceutical interventions in hub-based networks and the need to target the hub itself regardless of the outbreak's point of origin.

We evaluated source-city detection feasibility in the hub-satellite system as a function of transport-flow magnitude and daily testing fraction. For Moscow-region-like flows, the maximum achievable accuracy is about 78% when testing 10% of the population per day; reducing flows drives accuracy toward 100%. Accuracy heatmaps (testing fraction vs. epidemic day) reveal a parameter region where detection is impossible because too few infections are identified during that period. As transmissibility decreases, the boundary of this nondetection region shifts to later epidemic days, by roughly twofold at transmissibility equal to half the wild-type SARS-CoV-2 level. At the same time, increased relative variability in case counts at lower transmissibility degrades detection quality. This helps explain why, under Moscow-like flows, accuracy typically does not exceed about 70%, except at the 10% testing level where it can peak near 80%. Achieving 80% accuracy across testing levels appears feasible when flows are below roughly 3% of the real values at transmissibility equal to 0.5 of the wild-type SARS-CoV-2.

However, our study has several limitations related to the heterogeneous nature of traffic flows. First, we do not account for the coexistence of multiple transport modes. In real-world systems, mobility occurs across road, bus, rail, and air networks, each characterized by distinct topologies, temporal and spatial scales, and user populations, which may themselves differ in socioeconomic and behavioral attributes [6,8,22]. These differences can lead to substantially different patterns of connectivity and transmission potential across modes.

Second, travel is not instantaneous and may itself contribute to disease transmission through close contacts between passengers. The intensity and structure of such contacts depend strongly on the mode of transport, introducing additional heterogeneity that is not captured in our framework.

Third, the purpose of travel (e.g., commuting vs. tourism) introduces systematic variation in movement patterns and contact structures. Long-distance tourist trips often involve small, cohesive groups (e.g., families), whereas commuting typically exhibits strong spatiotemporal regularity, with repeated trips along fixed routes and stable contact patterns that occur primarily at destinations such as workplaces rather than during transit [26,39,40]. Neglecting these forms of regularity may lead to an overestimation of effective population mixing during travel and, consequently, to inflated estimates of transmission rates.

Finally, in systems with multiple transport modes, interventions targeting a single mode may induce adaptive redistribution of flows to alternative modes, thereby reducing the overall effectiveness of such measures. This substitution effect is not captured in the present model.

In future work, we plan to address these limitations by incorporating mode-specific mobility layers, explicitly modeling transmission within transport-specific contact networks, and accounting for the regularity and purpose of individual movement patterns. Achieving this level of detail, particularly at the country scale, will require high-resolution mobility data, such as mobile phone records or detailed travel surveys [41]. Developing robust methods for integrating such data while maintaining model tractability remains an important direction for further research.

## Methods

### Covasim

Covasim [30] is an open-source agent-based model that incorporates realistic population structure, disease progression, and interventions. In Covasim, each agent (one person or a group of people, depending on the population scale) can be

in one of these states: susceptible, exposed, infectious (divided into asymptomatic, presymptomatic, mild, severe, critical), recovered, and dead. The duration of each phase of the disease is determined individually for each agent from a lognormal distribution with parameters taken from COVID-19 studies. The probabilities of developing symptoms, severe symptoms, developing into a critical case, or dying, relative susceptibility vary for agents depending on their age, and these data are also taken from COVID-19 studies.

Agents are grouped into a social network containing households, schools, workplaces, and community contacts built using age-specific demographic data.

Transmission of infection can occur when there is contact between an infected and susceptible agent. The probability of transmission depends on the type of contact (household, workplace, school, community contact) and can be modified by interventions such as social distancing. In addition to varying susceptibility to infection depending on the age of the agent and the probability of transmission through contact, the infectiousness of agents is also individual and is determined by the distribution of viral load. The time dependence of the viral load in Covasim consists of two stages: an initial short stage with a constant high viral load, followed by a longer stage with a constant lower viral load. The viral load adjusts the probability of transmission through contact on a daily basis throughout the entire period of the agent's infectivity.

Each Covasim run consists of several stages. First, a simulation object is created and all necessary parameters are loaded. Then, agents are created according to age distribution data and connected into a contact network according to the selected method of synthetic population generation. Then, in a cycle at each time step, the population is scaled, i.e., dynamic rescaling, (if necessary to improve performance); health system constraints are applied; agent states are updated, including those related to the development of infection; random agents are infected (importation events); interventions are applied; the probabilities of further infection through the contact network are calculated; and the resulting metrics are calculated.

**transCovasim**

transCovasim is a framework for studying human mobility in agent-based models of epidemic spread. It organizes the transfer of agents between parallel simulations of the base agent-based model while preserving the initial social network structure within each population. The interaction between tourists and city residents is implemented using a separate social layer, and four additional agent attributes are added to ensure the movement of agents. transCovasim follows the same rules of infection transmission dynamics as the base model. In this work, the Covasim model [30] was taken as the base agent-based model.

The transCovasim model makes the following assumptions:

1. Agents move instantaneously between cities;

2. Agents in severe, critical, or dead states do not travel;

3. No infections occur during the instantaneous movement step;

4. Travelers are sampled uniformly at random from the origin population;

5. Upon arrival, travelers form random contacts within the destination city.

During initialization, each city is assigned a traveler-capacity equal to twice the expected number of travelers in that city, so, by default, the capacity is not saturated. Potential traveler agents are preallocated to each city as "absent" so that array sizes remain fixed during simulation, preserving performance, and they are configured to participate in social interactions per the specified parameters.

After initialization, the model proceeds through the main simulation loop. At each time step, a random set of traveler agents is drawn in each city according to the prescribed flows. These travelers are temporarily removed from their home

city's contact network and inserted into the destination city's network, where they engage in a fixed number of random contacts. After the trip duration elapses, the process is reversed. Agents in severe, critical, or dead states do not travel: if selected to travel, they remain in their home city; if they enter such a state while away, they return to the home city only after recovery. The core algorithms in Covasim's simulation loop are implemented as highly optimized operations on 32-bit Numba arrays [42]. For efficiency, agents are not instantiated as individual objects but as slices across state arrays.

To enable transport flows, we augmented Covasim's state-array schema with four agent-level attributes:

1. `trueId`: original identifier in the home city, used to restore identity upon return;

2. `inCity`: boolean flag indicating whether the agent is currently present in the city;

3. `restInAnotherCityDays`: number of days remaining in the current stay outside the home city;

4. `ownCity`: identifier of the agent's home city.

In the current model, the following traffic-flow parameters are configurable:

1. `adjacencyMatrix`: city-city flow matrix; each entry is the daily fraction of the origin city's population traveling to the destination;

2. `timeRelax`: mean trip duration in days (default: 7);

3. `interventionData`: transport-intervention spec; a dictionary indicating which links are restricted, the start day, duration, and the reduction factor applied to flows;

4. `contactCount`: mean number of contacts per traveler per day with residents of the destination city (default: 40);

5. `beta`: multiplier for transmission probability on the traveler layer (default: 0.3).

A detailed description of the algorithms is provided in the supplementary material (S1 File).

## Model validation

Covasim disease progressing parameters such as relative susceptibility to infection, probabilities of developing symptoms, severe symptoms, critical case, infection fatality ratio as well as parameters for default viral load distribution were validated from model fits to data on numbers of cases, numbers of people hospitalized and in intensive care, and numbers of deaths from Washington and Oregon states, USA and international estimates [30]. Moreover, Kerr et al. [30] showed that the Covasim model is able to reproduce the three distinct infection waves and temporal changes in test positivity rates during calibration period in King County, Washington, USA. During the prediction period, the model demonstrated correct prediction of the trend in cases and underestimated the number of deaths, however the data was still in the 80% forecast interval.

In the transCovasim model, we additionally validated the algorithms underlying human mobility: we checked the constancy of the total number of agents in simulations, the restriction on movement for agents in severe, critical, and dead states, the consistency of actual flows between cities with those specified in the parameters, and the return of tourists to their city of origin after the end of their trip. In addition, for each simulation, we added 15 simulation steps before seeding the infection to achieve a steady state in simulations where multidirectional transport flows are not balanced, and demonstrated the sufficiency of this period (S3 Fig).

## Design of the experiments

Our work includes a study of four scenarios of epidemiological computational experiments. We constructed these scenarios by gradually increasing the complexity of the model and approaching the main task of detecting the outbreak's origin. The first

scenario (two-city transport model with equal agent counts) is a minimal model for determining the rate of infection spread between cities in the absence of asymmetry in flows, with the aim of examining the influence of flow magnitude and infection transmissibility. The second scenario (two-city transport model with unequal populations) complicates the model by introducing asymmetry in transport flows to analyze the impact of flows in different directions on key epidemiological metrics (cumulative and peak infections, peak day of infections). The third scenario (hub-and-satellite transport model) introduces a more realistic transport model containing a large central city and four smaller satellite cities connected to the central city by large commuting flows. This scenario allows us to investigate how the city where the epidemic began (hub or satellite) affects the course of the epidemic and the effectiveness of transport flow restrictions. The fourth scenario (detecting the outbreak's origin in the hub-satellite model) focuses on the main task of our work, which is to identify the outbreak city based on time series of the number of infections in cities in the hub-satellite model with different transport flow multipliers and population testing probabilities.

Computational experiments were conducted in Python. For each parameter set, we ran 150 simulation replicates with different random seeds. Metrics are reported as the mean across 150 runs, with standard deviations indicating variability.

In the two-city experiments, the mean stay in the destination city was 7 days, the number of contacts was 40, and the transmissibility multiplier for travelers was 0.3 (as for random contacts in [30]). For the identical-cities scenario, we excluded outliers before further analysis, specifically, simulations in which the epidemic failed to start or did not spread to the neighboring city.

In addition to such simulations for scenarios with an unequal number of agents in cities, as well as in all subsequent experiments, simulations in which the epidemic spread only to a small part of the population of one of the cities were also considered as outliers. All metrics of such simulations for this scenario were replaced with average values from simulations with other random seeds (outliers accounted for about 0.5% of all simulations) rather than excluded to ensure the possibility of calculating sensitivity indices. In this scenario, Sobol's first indices were used to assess the sensitivity of epidemiological indicators to traffic flows, with flows varying independently in the range from $10^{-5}$ to $10^{-3}$ in 4,096-sample Saltelli design [33]. Sobol' sequences and first-order sensitivity indices were computed using SALib [43,44].

In the hub-satellite experiments calibrated to Moscow and the Moscow region, the region was partitioned into four satellite cities. Hub-to-satellite population ratio matches that of Moscow and the Moscow region, while the total number of agents is set to one-twentieth of the real population to reduce computation time. Bidirectional flows between the hub and satellites were set from mobile-operator data [34]. Inter-satellite flows were estimated as proportional to city populations and inversely proportional to the square of inter-city distance [35–37]. For simplicity, we used a square geometry with the hub at the center: the distance between neighboring satellites is $\sqrt{2}$ times the hub-satellite distance, and between opposite satellites is 2 times the hub-satellite distance. Trip duration was 1 day, the number of contacts was 40, and the traveler-layer transmission multiplier was 0.6, consistent with short commuter trips to workplaces in neighboring cities.

Restriction strategies for hub-satellite experiments were designed as follows: from 31 January 2020 to March 2020 actual mobility from Wuhan to other Chinese cities was 15–20 times smaller than expected volumes [45], number of international tourists decreased by 58–78% due to COVID-19 [46]. We therefore considered 10-fold restriction in flows as a realistic strict scenario, and 100-fold restriction as an extreme scenario. Reducing flows only on links incident to the outbreak city is a more targeted strategy, while reducing all flows is a more general strategy that does not require identifying the outbreak city. We introduced restrictions at various times to evaluate the importance of timing for intervention effectiveness.

In the detection experiments, testing was simulated by sampling detected cases from a binomial distribution whose success probability equaled the true proportion of new infections.

## Supporting information

**S1 Fig. Epidemic spread in a two-city system with identical populations.** A–B: Heatmaps of the mean offset in the peak-infection day of the destination city relative to the origin city (A) and of the t-statistic comparing peak days (B), based on the transport flow between cities (fraction of the population per day) at different infectiousnesses (fraction of the infectiousness

of the wild variant SARS-CoV-2). C: Histograms of the distribution of the peak infection day in the city of origin (blue) and in a neighboring city (orange) at different levels of traffic flow between cities (columns) and infectiousnesses (rows). D: Phase portrait of the two-city system at different traffic flows between them at the infectiousness of the corresponding wild variant SARS-CoV-2. E: Dependence of the t-statistic when comparing the days of peak infections in cities in the system on the traffic flow between cities at different infectiousnesses. Each panel shows the result of 150 simulations after removing outliers.
(TIF)

**S2 Fig. Detecting the outbreak's origin in the hub-satellite model.** A–C: Heatmaps of origin-detection accuracy versus epidemic day for systems with traffic flows 3.16 (A), 10 (B), and 31.6 (C) times smaller than those of Moscow and the Moscow region. Testing is simulated by sampling detected infections from a binomial distribution. Infectiousness set to the wild-type SARS-CoV-2 level.
(TIF)

**S3 Fig. Stabilization of the number of agents in cities.** A–D: Time series of the deviation of the number of agents in each city during the stabilization period 15 steps before seeding infection from the median value for all time steps of a specific simulation for two-city system with identical populations (A), two-city system with unequal populations (B), hub-satellite system (C), and hub-satellite system with outbreak origin detection (D).
(TIF)

**S1 File. Descriptions of the algorithms underlying the transCovasim model.**
(PDF)

## Author contributions

**Conceptualization:** Konstantin A. Klochkov, Alexander I. Manolov.

**Data curation:** Konstantin A. Klochkov, Ivan E. Kozlov.

**Formal analysis:** Konstantin A. Klochkov, Alexander I. Manolov.

**Funding acquisition:** Elena N. Ilina.

**Investigation:** Konstantin A. Klochkov.

**Methodology:** Konstantin A. Klochkov, Ivan E. Kozlov, Alexander I. Manolov.

**Project administration:** Elena N. Ilina, Alexander I. Manolov.

**Resources:** Elena N. Ilina.

**Software:** Konstantin A. Klochkov, Ivan E. Kozlov.

**Supervision:** Elena N. Ilina.

**Validation:** Konstantin A. Klochkov, Ivan E. Kozlov.

**Visualization:** Konstantin A. Klochkov.

**Writing – original draft:** Konstantin A. Klochkov.

**Writing – review & editing:** Konstantin A. Klochkov, Alexander I. Manolov.

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
