## [Decision Letter · Decision Letter 0]

5 Jan 2026

PCOMPBIOL-D-25-02107

Transportation flows and outbreak origins in epidemic spread: Insights from agent-based modeling

PLOS Computational Biology

Dear Dr. Klochkov,

Thank you for submitting your manuscript to PLOS Computational Biology. After careful consideration, we feel that it has merit but does not fully meet PLOS Computational Biology's publication criteria as it currently stands. Therefore, we invite you to submit a revised version of the manuscript that addresses the points raised during the review process.

We look forward to receiving your revised manuscript.

Kind regards,

Shi Zhao

Academic Editor

PLOS Computational Biology

Denise Kühnert

Section Editor

PLOS Computational Biology

**Journal Requirements:**

At this stage, the following Authors/Authors require contributions: Konstantin A. Klochkov, Elena N. Ilina, Ivan E. Kozlov, and Alexander I. Manolov. Please ensure that the full contributions of each author are acknowledged in the "Add/Edit/Remove Authors" section of our submission form.

3) Some material included in your submission may be copyrighted. According to PLOSu2019s copyright policy, authors who use figures or other material (e.g., graphics, clipart, maps) from another author or copyright holder must demonstrate or obtain permission to publish this material under the Creative Commons Attribution 4.0 International (CC BY 4.0) License used by PLOS journals. Please closely review the details of PLOSu2019s copyright requirements here: PLOS Licenses and Copyright. If you need to request permissions from a copyright holder, you may use PLOS's Copyright Content Permission form.

Potential Copyright Issues:

i) Figures 1A, 2A, 2B, and 3A. Please confirm whether you drew the images / clip-art within the figure panels by hand. If you did not draw the images, please provide (a) a link to the source of the images or icons and their license / terms of use; or (b) written permission from the copyright holder to publish the images or icons under our CC BY 4.0 license. Alternatively, you may replace the images with open source alternatives. See these open source resources you may use to replace images / clip-art:

4) Please amend your detailed Financial Disclosure statement. This is published with the article. It must therefore be completed in full sentences and contain the exact wording you wish to be published.

**Reviewers' comments:**

Reviewer's Responses to Questions

**Comments to the Authors:**

Reviewer #1: It is meaningful to investigate how mobility influence disease spreading and ABM is appropriate. However, the rules of transmission dynamics in this study is not clear enough.

The reviewer has some questions and suggestions as follows:

1. I think mobility and transportation flow are in different scopes. Please check the difference between these terms. The definition of inbound and outbound flows are not clear as well. It is suggested to rename your title.

Introduction:

2. The existing structure has listed relevant evidence of different epidemics and shown the connection between mobility and disease spreading. However, this part is too redundant. How about different methodologies that help investigate how mobility influence disease spreading? Overall, please state the research gap of developing an agent-based transCovasim model in this study.

Methods:

3. Please state your transCovasim model by introducing three components 1) agent; 2) environment; 3) interaction rules, which is widely used in the filed of ABM. Please add a technical framework of ABM.

4. the rule of transmission dynamics of deases should be better explained in the transCovasim model. Have you considered the heterougeneity brought by mobility through different transportation modes? If no, why you select ABM as your method.

Design of the experiments:

5. This study has three scenarios: 1) two-city transport model with equal/unequal agent counts; 2) hub-and-satellite transport model and 3) detecting the outbreak’s origin in the hub-satellite model. Please state the reason why you designed these experiments.

Results:

6. how about the validation of your model?

7. how did you design restriction strategies?

Reviewer #2: This paper presents and analyses transCovasim, an extension of the Covasim agent-based model of COVID-19. Whereas Covasim models a single population, this extension enables the authors to study mixing between two identical populations, unequal populations, and a hub-and-spoke configuration. Results show how population sizes and migration flows affect disease burden, and indicate how interventions could mitigate/delay the peak and total burden.

Overall, the manuscript is clear and well written. The results are a bit dense and detailed, and some readers may get lost in the details of origin and neighbor nodes. The figures support the main results presented by the authors.

I have a few technical questions for author consideration:

 Dynamic rescaling. Covasim has a `rescale` parameter that is set to True by default. This parameter configures dynamic rescaling. In Covasim, there is only one population, and all agents have the same scaling factor. With dynamic rescaling, the effective scale of agents increases over time. Conceptually, this process represents "zooming out" from an initial small seed area to the full population being simulated.

The implementation here is quite clever. While Covasim runs a single process, the authors have extended the MultiSim capabilities using multiprocessing "barrier" and "lock" to synchronize. However, with this modification, there are multiple instances of the simulation running in lockstep, but each instance may have a different population scaling factor. An agent migrating from one city to the next may have a different scale factor.

The scale factor increases following seeding, but quickly stabilizes at a maximum value -- so it might be a non-issue. Please check, and consider rerunning the simulation experiments with rescaling disabled.

 I wonder about population sizes. Each node starts with a specified population size, but the number of agents in each city will change dynamically as migration unfolds (unless balanced?).

• If not balanced, consider adding a supplementary figure showing how the population in each city changes throughout the course of a simulation.

• Worth checking that the upper limit of population is not reached.

• Additionally, it might be worth letting between-city mobility (population sizes) reach steady state before seeding infection.

• Finally, ensure that key statistics like incidence and prevalence are calculated correctly with dynamic city sizes.

 The mixing structure within each city is in fact quite detailed in these simulations. I did not realize this until looking at the source code. The populations are built using the SynthPops package configured for Seattle (for which rich microdata are available). It might be worth a sentence or two in the methods section describing that agents have static home/school/work/community connections that are not affected by between-city migration.

 The approach to moving agents between cities, each running in a separate simulation, is already quite novel! I might suggest that the authors describe that agents are physically moved, not the a rewiring of the network. This pattern could be reused by others, so highlighting code availability and key changes from Covasim would be welcome.

 In the hub-and-spoke source identification, are symmetries holding back results. Because all satellite communities are equal in size and connectivity to Moscow, it would seem challenging to identify the source. Would the source be more identifiable if node sizes and/or connection weights were heterogeneous?

Overall, I find this work interesting and potentially useful if oriented towards a "next pandemic" or disease X. The conclusions support interventions affecting population mobility. In practice, my understanding is that mobility restrictions have demonstrated mixed impact. The literature review focuses attention to the early outbreak in China and other simulation papers, but a few more examples here to illustrate where such interventions have worked/failed may help readers connect this work to practical policy.

**Have the authors made all data and (if applicable) computational code underlying the findings in their manuscript fully available?**

Reviewer #1: Yes

Reviewer #2: Yes

PLOS authors have the option to publish the peer review history of their article (what does this mean?). If published, this will include your full peer review and any attached files.

Reviewer #1: No

Reviewer #2: No

**Figure resubmission:**
---

## [Decision Letter · Decision Letter 1]

15 Apr 2026

PCOMPBIOL-D-25-02107R1

Human mobility and outbreak origins in epidemic spread: Insights from agent-based modeling

PLOS Computational Biology

Dear Dr. Klochkov,

Thank you for submitting your manuscript to PLOS Computational Biology. After careful consideration, we feel that it has merit but does not fully meet PLOS Computational Biology's publication criteria as it currently stands. Therefore, we invite you to submit a revised version of the manuscript that addresses the points raised during the review process.

We look forward to receiving your revised manuscript.

Kind regards,

Shi Zhao

Academic Editor

PLOS Computational Biology

Denise Kühnert

Section Editor

PLOS Computational Biology

**Journal Requirements:**

1) Please amend your detailed Financial Disclosure statement. This is published with the article. It must therefore be completed in full sentences and contain the exact wording you wish to be published.

1) State the initials, alongside each funding source, of each author to receive each grant. For example: "This work was supported by the National Institutes of Health (####### to AM; ###### to CJ) and the National Science Foundation (###### to AM).".

**Reviewers' comments:**

Reviewer's Responses to Questions

**Comments to the Authors:**

Reviewer #1: The authors have addressed most of my concerns in this revision.

However, I am still worried about model heterogeneity.

Line 301-310：Several limitations mentioned here are all related to model heterogeneity, but they are described less specific. Could you please elaborate on the heterogeneity issues brought by different situations that you mentioned here, and add future research directions or plans to better list appropriate solutions?

**Have the authors made all data and (if applicable) computational code underlying the findings in their manuscript fully available?**

Reviewer #1: Yes

PLOS authors have the option to publish the peer review history of their article (what does this mean?). If published, this will include your full peer review and any attached files.

**Do you want your identity to be public for this peer review?** For information about this choice, including consent withdrawal, please see our Privacy Policy.

Reviewer #1: No

**Figure resubmission:**
---

## [Editor Report · Decision Letter 2]

28 Apr 2026

Dear Mr. Klochkov,

We are pleased to inform you that your manuscript 'Human mobility and outbreak origins in epidemic spread: Insights from agent-based modeling' has been provisionally accepted for publication in PLOS Computational Biology.

Best regards,

Shi Zhao

Academic Editor

PLOS Computational Biology

Denise Kühnert

Section Editor

PLOS Computational Biology

---

## [Editor Report · Acceptance letter]

PCOMPBIOL-D-25-02107R2

Human mobility and outbreak origins in epidemic spread: Insights from agent-based modeling

Dear Dr Klochkov,

I am pleased to inform you that your manuscript has been formally accepted for publication in PLOS Computational Biology. Your manuscript is now with our production department and you will be notified of the publication date in due course.

With kind regards,

Zsofia Freund
